# Performance of 3D-Printed Beams and Slabs Using Self-Sensing Cementitious Composites and DIC Method

**DOI:** 10.3390/s23208486

**Published:** 2023-10-16

**Authors:** Zhuming Li, Farhad Aslani

**Affiliations:** Materials and Structures Innovation Group, School of Engineering, The University of Western Australia, Crawley, WA 6009, Australia; 23092167@student.uwa.edu.au

**Keywords:** self-sensing, piezoresistivity, 3D printing, beams, slabs, digital image correlation

## Abstract

This paper aims to explore the structural performance of 3D-printed and casted cement-based steel-reinforced concrete beams and one-way slabs incorporating short carbon fibre and activated carbon powder, which have been shown to enhance concrete’s flexural strength and reduce its electrical resistivity. The samples are cast and printed in 250 × 325 × 3500 mm beams and 150 × 400 × 3500 mm one-way slabs and mechanical, electrical, and piezoresistivity properties were measured. This length of beams and one-way slabs with rebars have been considered as they can magnify the flexural and cracking behaviour and make them easier to be monitored and analysed. The samples were loaded up to 80% of maximum stress. Crack propagation and strain was assessed using the 2D digital image correlation (DIC) method. The results compared samples under continuously increasing loads between 3D-printed and cast samples. The 3D-printed composites had a better piezoresistive response due to the enhanced anisotropic behaviour. DIC analysis illustrated similar results among different samples, while 3D-printed blocks had lower cracking performance due to the horizontal case fracture in lower stress.

## 1. Introduction

Because of its abundant resources, affordability, high strength, excellent durability, and various other attributes, concrete has emerged as the most extensively utilised construction material. It has found successful applications in diverse engineering sectors such as residential buildings, road infrastructure, bridges, dams, and more [1]. During the service life of concrete structures, various factors like shrinkage, expansion, external loads, and complex environmental conditions can lead to volumetric deformations, resulting in the formation and propagation of internal cracks within the concrete. These cracks directly undermine the performance, longevity, and even the safety and reliability of concrete structures. Hence, there is a critical need for research in the field of non-destructive structural health monitoring (SHM). Traditionally, SHM has relied on sensors, strain gauges, and ultrasonic methods to measure strain and deflection in concrete structures. However, these approaches have limitations. Sensors often suffer from inadequate sensitivity and durability, while ultrasonic instruments are prohibitively expensive for routine use in ordinary conditions [2,3].

The incorporation of digital image correlation (DIC) offers a valuable means to detect structural cracks and strain. When compared to alternative testing methods, DIC stands out due to its numerous advantages, including real-time measurement, non-destructive monitoring, high precision, and the capability to provide comprehensive strain information across the entire field. Since the inception of the DIC method [4], it has garnered widespread attention, prompting extensive research efforts aimed at expanding and enhancing its utility in deformation analysis. Initially, DIC was employed to monitor the crack propagation in conventional metal materials [5], contributing significantly to fracture mechanics research. Subsequently, it found application in the field of civil engineering, where DIC, in conjunction with high-speed image acquisition equipment, was utilised to investigate crack growth under impact loads [6]. This approach has proven to be highly effective in tracking the evolution of crack propagation within reinforced concrete (RC) structures.

During the experiment, DIC proved invaluable for tracking the complete evolution of displacement and strain fields, spanning from the initiation of cracks to their propagation and eventual failure [7]. DIC not only aids in pinpointing the initial crack’s location and direction on the concrete surface but also facilitates the precise monitoring and evaluation of crack propagation and profiles [8]. Furthermore, DIC enables the measurement of displacement field changes and thermodynamic properties in specimens subjected to constant amplitude loads [9]. Beyond these applications, DIC finds utility in examining the deformation field on concrete surfaces, allowing for the accurate prediction of potential crack paths under fatigue loads through the analysis of strain fields. This capability aids in studying crack propagation on concrete surfaces, and subsequently predicting the service life of the material [10]. DIC also comes into play when studying the multidirectional crack propagation processes in concrete material specimens [11].

The research encompassed real-time monitoring of the compressive fracture processes in cellular concrete, ordinary strength concrete, and high-strength concrete [12]. Additionally, it delved into the elastic behaviour of fibre-reinforced cement-based materials [13]. Investigations extended to the bending and cracking characteristics of conventional concrete beams, high-strength concrete beams, and high-strength fibre concrete beams, with a focus on accurate initial crack detection and tracking the evolution of cracks and strains [14]. The study also examined the fracture behaviour of concrete under cyclic fatigue loading conditions [15,16]. To gain insights into the fracture properties of concrete and fibre-reinforced cement-based materials, researchers employed acoustic emission technology [17,18]. Furthermore, X-ray tomography was utilised to explore the splitting mechanics and fracture characteristics of plain concrete, recycled steel fibre concrete, and industrial steel fibre-reinforced concrete. Within this framework, DIC was instrumental in scrutinizing the development of the fracture process zone on the sample surface and the progression of internal cracks within concrete, using X-ray tomography as a complementary tool [19]. The investigation encompassed the cracking phenomenon observed in mortar subjected to dry conditions [20]. Additionally, the study focused on plastic shrinkage and cracking in cement-based materials, offering an in-depth analysis of the plastic shrinkage and cracking process through a series of strain contour assessments [21].

The paper is structured as follows: 1. The uses of DIC are summarised; it has emerged as an alternative to traditional contact measurements for evaluating a wide range of mechanical properties in concrete materials, including stress intensity, residual stress, and Young’s modulus. 2. DIC is applied to investigate the deformation patterns in concrete materials subjected to different mechanical loads, leveraging its capacity to easily detect significant deformations. 3. The research then investigates the formation and spreading of cracks in concrete, often in conjunction with complementary testing methods. And 4. DIC is utilised for the real-time monitoring of volume deformations and the localized propagation of cracks within concrete structures.

Material-extrusion-based 3D Printing, also referred to as additive manufacturing or rapid prototyping, represents a method for crafting intricate geometric shapes and structures based on computer-generated models [22]. This technology not only offers high automation and manufacturing precision, but also leads to reduced labour requirements, shorter project timelines, and cost savings [23,24]. In the realm of 3D printing cementitious composites, a layered structure is created by extruding filaments through a nozzle, with the embedded fibres aligning along the printing path. Consequently, extrusion-based 3D printing has enhanced the mechanical attributes of cementitious composites, and holds the potential to improve their electrical and piezoresistive properties. Recognizing these advantages, 3D printing technology holds substantial promise within the future construction industry. Therefore, it becomes imperative to explore the realm of 3D printing self-sensing cementitious composites, aiming to enable the structural health monitoring (SHM) of upcoming infrastructure projects.

This research integrates self-sensing cement composites with 3D printing technology to produce self-sensing cement-based composites. The study specifically investigates compositions comprising 0.7 wt% carbon fibre (CF) and 0.25 wt% activated carbon powder (ACP), as these formulations demonstrated the most favourable piezoresistive response, as confirmed through slump and flow table tests [25]. The study involved the assessment of mechanical, resistivity, and piezoresistive characteristics of 3D-printed samples in two orientations: one parallel to the printing direction and the other perpendicular to it. Additionally, die-cast specimens were fabricated for a direct comparative analysis. Furthermore, the piezoresistivity technique was employed to evaluate enhancements in linearity, repeatability, and signal quality when testing 3D-printed reinforced cementitious composites incorporating long carbon fibres.

## 2. Materials and Methods

### 2.1. Materials

The cementitious composite formulations chosen for this study have been previously validated for their favourable piezoresistive properties [25,26,27]. When considering carbon-fibre-reinforced concrete structures, it is important to note that the mechanical properties of the constituent materials, as well as the size and reinforcement of the specimens, directly influence the mechanical parameters. In this research, the reinforced cementitious composites were crafted using general-purpose Portland cement, ground granulated blast furnace slag (GGBFS), and densified silica fume (SF).

#### 2.1.1. Cementitious and Supplementary Cementitious Materials

General purpose cement AS 3972 Type GP, manufactured by Cockburn Cement Limited, was selected for use in this study. The chemical compositions and physical properties of both the cement and ground granulated blast furnace slag (GGBFS) are detailed in Table 1, Table 2, Table 3 and Table 4. The densified silica fume (SF) utilized in this research, provided by Microsilica Pty Ltd. (Wellesley, WA, Australia), conforms to AS 3582 part 3 standards. Table 5 and Table 6 present the chemical compositions and physical properties of SF. The incorporation of densified silica fume plays a crucial role in reducing van der Waals forces between fillers, thus enhancing the microstructure density and the even distribution of fillers within the matrix.

#### 2.1.2. Carbon Fibre and Activated Carbon Powder

The functional fillers employed to confer piezoresistive characteristics comprise CF and ACP. Unsized polyacrylonitrile CF was selected due to its exceptional tensile strength, compressive strength, and electrical attributes, as previously validated [27]. ACP, on the other hand, is incorporated with the specific objective of enhancing the piezoresistive properties within self-sensing reinforced cementitious composites. Table 7 and Table 8 present the CF and ACP properties.

#### 2.1.3. Admixtures

Master Glenium SKY 8708, obtained from MB Solutions Australia Pty Ltd. (Wellesley, WA, Australia), NSW, was utilized as a High-Range Water Reducing Agent (HRWRA) compliant with the requirements for High-Range Water Reducers specified in AS 1478.1-2000. The application of HRWRA serves multiple purposes, including enhancing the dispersion of fillers, improving workability, reducing unit water consumption, enhancing the fluidity of the concrete mixture, and cost-effectiveness. Simultaneously, Master Matrix 362 was introduced as a Viscosity-Modifying Agent (VMA) to enhance cement viscosity and control rheological properties. VMA plays a pivotal role in facilitating extrudability by mitigating the bleeding effect typically associated with cementitious materials [25]. Notably, VMA enhances the yield stress of mortar at low dosages through a bridging effect, while having minimal impact on plastic viscosity. However, when used at high dosages, the bridging effect of VMA diminishes, leading to a significant increase in the viscosity of the liquid phase. This, in turn, reduces the enhanced effect on mortar yield stress while substantially elevating the plastic viscosity of the mortar [28].

#### 2.1.4. Mix Compositions

Table 9 provides a comprehensive listing of all components utilized in the composition of cementitious composites for both the 3D-printed self-sensing samples and the cast specimens. The optimal water-to-binder ratio, set at 0.325, was chosen to achieve optimal flowability, particularly in conjunction with CF and ACF. These additives were introduced to align with the requirements of the extrusion process, considering their influence on printability, as demonstrated through previous investigations [25].

### 2.2. Beams and Slabs Preparation

The dimensions for the beams and slabs were derived from Aslani’s work [29]. The beam section measures 250 mm × 325 mm × 3500 mm and incorporates 2 × 16 mm steel rebars. In contrast, the slab section measures 150 mm × 400 mm × 3500 mm and contains 4 × 12 mm steel rebars. For each structural element, two specimens were prepared: one cast, and the other assembled using 3D-printed blocks. The specific dimensions and arrangement of rebars are illustrated in Figure 1.

A desktop concrete 3D printer was employed to manufacture the concrete components through an extrusion process. Within the printer container, the cementitious composites were extruded along a predefined path specified by the g-code program. Concurrently, four copper meshes were strategically inserted at designated locations to serve as electrodes for measuring electrical properties. After a curing period of 28 days, eight printed blocks were prepared to serve as the outer shell for the 3.5 m beams and slabs, as depicted in Figure 2 and Figure 3. In contrast, traditional mould cast samples were also produced alongside the 3D-printed specimens. Following the same 28-day curing period, all samples, whether 3D-printed or cast, were diligently covered with plastic film and subjected to daily curing procedures to mitigate the risk of shrinkage and cracks.

### 2.3. DIC Analysis

Strain Calculation and Contour

The primary elements comprising the DIC experimental configuration for this flexural experiment encompass a high-resolution CMOS camera and a computer, depicted in Figure 4. In the validation test for displacement accuracy involving auxiliary components, a linear variable differential transformer (LVDT) was employed to measure displacement, with the adjusted LVDT reading considered as the reference value. Concurrently, strain data was captured through an integrated strain gauge to corroborate the precision of DIC measurements. To augment image quality, an LED cold light source was used to compensate for ambient lighting conditions.

In the DIC experimental setup, to ensure efficient data transmission and computational processes, the image acquisition and associated calculations are typically executed as separate procedures. For data post-processing in this study, we employed the open-source DIC calculation software MultiDIC 6.0.4., which is MATLAB-based and is represented by its operational interface, displayed in Figure 5. This software amalgamates essential functionalities, encompassing calibration and pertinent calculations. Furthermore, this aspect of the research utilized Adobe Lightroom for image processing. Lightroom offers a comprehensive array of commonly employed image-processing techniques and boasts robust capabilities in this domain.

Following the experimental setup, the AMSLER system initiated the loading of specimens, while, concurrently, the DIC system commenced capturing images of the designated observation area. In the image processing phase, the appropriate calibration data were selected. Then, upon entering the calculation interface, the region of interest for analysis was delineated on the image. Subsequently, analysis parameters were configured, and a seed point within the calculation area was designated, enabling the commencement of seed point analysis. Throughout the experiment, the seed point selection typically entailed identifying distinct features or striations within the calculation area as the initial point for analysis. Figure 6 illustrates the test assessing the damage and failure behaviour of identical cementitious composites when subjected to both flexural and cyclic loads. During the experiment, an initial observation of structural strain under applied force was conducted.

### 2.4. Measurement of Crack Depth

In this experimental context, the term “crack depth” refers to the vertical distance extending from the tip of the crack to the base of the strengthened beam. Consequently, this experimental approach entails two essential elements for gauging the crack height. The first involves quantifying the distance from a specific point (the crack tip) to a straight line (the bottom of the strengthened beam). The second entails accurately determining the location of the crack tip. In Figure 7, it is evident that the lowermost portion of the beam, which corresponds to the peripheral area within the image, cannot be entirely encompassed within the DIC calculation area. Consequently, when outlining the calculation area, deliberate measures were taken to ensure that its lower boundary remains parallel to the base of the beam.

Additionally, it is worth noting that in certain instances, the coordinate system used for calculations may differ from the actual spatial coordinate system, resulting in the origin of the coordinates occasionally appearing outside the designated calculation area. This discrepancy arises from the calculation methodology and the displacement of the beam, rather than stemming from calculation errors. In such cases, the change in relative crack height can be effectively assessed by drawing a vertical line in space that aligns with a fixed parallel reference line, as illustrated in Figure 7. This methodology was employed for measuring the crack depth of the specimens within this experiment.

### 2.5. Mechanical Properties

As depicted in Figure 8, the laboratory was equipped with a functional platform designed to seamlessly integrate with the AMSLER loading machine, which boasts a loading rate of 0.001 mm/s and a loading capacity of 2000 kN. The AMSLER loading machine, an electrohydraulic universal testing apparatus, is specifically designed to conduct high-load tests under external control inputs, facilitating comprehensive large-scale experiments. In this study, all specimens were placed in a simple supported configuration over a 3.5 m span, and they were subjected to failure tests. The objective was to analyse the distribution and extent of both primary and secondary cracking under static loading conditions. Two equal point loads were applied at the third points along the span, with all tests conducted on specimens aged greater than 28 days.

For the four-point bending test, the loading points were positioned 500 mm away from the centre. The midpoint deflection was continuously monitored using a linear variable differential transformer (LVDT). Furthermore, strain gauges were affixed to each of the steel reinforcements at intervals of 200 mm, as depicted in Figure 9. To capture strain data accurately, digital instrument amplifiers were employed to process and analyse the signals detected by the strain gauges.

### 2.6. Electrical Resistivity

During the printing process, four copper meshes were incorporated into the block to serve as embedded electrodes for the purpose of electrical performance evaluation. These meshes were strategically positioned at a distance of 900 mm from the centre, with a spacing of 150 mm between them.

As illustrated in Figure 10, the prepared sample featured the inclusion of four copper meshes, external probes, and embedded electrodes. The external probes facilitated current input and voltage output measurements, allowing the determination of resistivity along a path parallel to the printed trajectory. To establish a basis for comparison, the resistivity of the mould-cast sample was also assessed. It is worth noting that, for the sake of simplicity, all prepared samples were subjected to DC (Direct Current) measurement rather than AC (Alternating Current). A digital multimeter, specifically the Keithley 2100, was employed for measuring DC resistance, with a trigger interval of 25 ms.

### 2.7. Piezoresistivity

This study entailed the evaluation of piezoresistive properties in both 3D-printed and mould-cast samples. The flexural load applied aligned with the printing path direction in the case of printed samples. To gauge the impact of applied loading on fractional change in resistivity (FCR), DC resistance measurements were conducted, employing a Keithley 2100 digital multimeter [25].

Equations (1) and (2) were used to evaluate the electrical resistivity (*ρ*) (Ω·cm) and the FCR (Δ*ρ*) (%) of the samples.
(1)$$\begin{array}{c} \rho =\frac{RA}{L} \end{array}$$
(2)$$\begin{array}{c} \Delta \rho =\frac{\rho_{m}-\rho_{o}}{\rho_{o}}\times 100\% \end{array}$$
where *R* is the electrical resistance from multimeter (Ω); *A* is the action area between the copper meshes and the cementitious composites (cm^2^); *L* is the distance between the inner two electrodes (cm); and $\rho_{m}$ and $\rho_{o}$ are the changing electrical resistivity and initial electrical resistivity (Ω·cm).

## 3. Results and Discussion

### 3.1. DIC Deformation Analysis under Flexural Loading

#### 3.1.1. 3D-Printed Concrete Beam Specimens

Figure 11 presents the load–time and load–deflection curves obtained from the experimental analysis of the 3D-printed beam. Notably, the curves during the initial loading phase exhibit transient non-linear characteristics. As the loading duration progressed, the specimen exhibited a stable linear curve, transitioning into an elastic behaviour approximately at the 2000 s mark. Subsequently, as the load reached its yield point, the specimen experienced a rapid onset of cracking. However, due to the reinforcement provided by the rebar, the sample did not fail immediately. Instead, it underwent plastic yielding and strain hardening, manifesting as a distinctive phase between point B and point C. Following this phase, the load experienced a rapid decline after reaching 58 mm, or approximately 3300 s, ultimately concluding the loading process.

Figure 12 illustrates the progression of the strain field computed from images captured at a frame rate of 0.5 Hz. In the initial stages of loading, the strain distribution within the specimen remained uniform. As the loading intensified, notable strain concentration became evident near the bottom of the sample, particularly in proximity to the peak load point (point B). Subsequently, when the load reached its pinnacle, the specimens exhibited visible cracking, with these cracks penetrating through the material. It is worth noting that, despite the presence of cracks, the test piece remained intact and continued to bear the applied load. Further analysis delved into the evolution of strain within the specimens as the cracks propagated. At 709.689 s, a zone characterized by high-amplitude deformation and strain values of approximately 0.05 emerged near the loading end. This deformation zone subsequently expanded towards the centre of the specimen. By 2401.864 s, the strain within this deformation concentration area increased in response to the growing load, and the extent of the strain concentration area continued to broaden. Eventually, the depth of this expansion area reached 223.3 mm, corresponding to 75% of the specimen’s overall height. At 3280.25 s, the deformation concentration area spanned the entire specimen, resulting in the failure of the specimen. Notably, during this process, horizontal cracks were observed on the printed shell, particularly at point B, from the formation of the strain concentration region to the ultimate failure of the specimen.

In regions of a continuous surface, displacements change in a continuous and uninterrupted manner. However, the presence of cracks in such areas disrupts this continuity, leading to displacement jumps or discontinuities. This fundamental criterion serves as the basis for crack identification through the DIC testing method. Consequently, the calculated displacement field clearly reveals the existence of cracks, as demonstrated in Figure 12. Figure 12 provides an insight into the horizontal displacement field of the printed beam during deflection loading and at the point of maximum deflection for the reinforced beam. Both horizontal displacement (dX) and vertical displacement (dY) can serve as indicators for identifying cracks. However, in the context of this experiment, the cracks predominantly exhibited a vertical orientation, and their propagation direction aligned with the vertical axis throughout the entire test. Consequently, all crack identifications within this study were based on the transverse direction.

As depicted in Figure 12, the specimen’s surface experienced multiple instances of cracking. These cracks were discernible within the calculated area of the specimen. Furthermore, the width of the cracking area serves as a visual indicator of the crack’s height. Notably, the primary crack identified in the fatigue test of the reinforcing beam was positioned centrally within the specimen. Figure 13 provides a magnified view of the crack surface, magnified 500 times, confirming the presence of cracks in the specified area. This observation substantiates the hypothesis outlined above.

#### 3.1.2. Cast Concrete Beam Specimens

To assess and contrast stress concentration within the cementitious composites of the printed specimen, a flexural loading test was conducted, alongside similar tests performed on cast beams. The load–time curves representing the outcomes of both sets of experiments are presented in Figure 14. Notably, these curves follow a consistent pattern: they exhibit a brief non-linear phase at the commencement of loading, subsequently maintaining a linear trajectory, albeit with the emergence of several minor cracks. Following point B in the loading process, the curves begin to exhibit fluctuations. It is at point C that the load experiences a precipitous decline, culminating in the eventual failure of the specimen.

Figure 15 provides a comprehensive depiction of the surface strain field’s evolution under the loading conditions applied to the cast beams. In the initial stages of loading, the cast concrete exhibited a relatively uniform distribution within the specimen’s plane, resulting in a corresponding evenness in strain distribution. Upon reaching the peak yield strength (point B), a notable concentration of strain emerged in proximity to the centre of the loading disc’s diameter, registering at approximately 0.009. Subsequently, this strain concentration rapidly escalated to around 0.017. Concurrently, the declining load signified the initiation of central crack propagation within the specimen. Beyond point B, the strain concentration area progressively extended from near the specimen’s centre towards both ends, accompanied by continuous crack development. Ultimately, at moment C, the crack successfully traversed the entire specimen, leading to its fragmentation and fracture. It is noteworthy that the deformation field exhibited a similar evolution pattern in another cast beam.

#### 3.1.3. 3D-Printed and Cast Concrete Slabs

To gain deeper insights into the impact of various specimen shapes on deformation evolution, a series of flexural tests was conducted on both printed and cast slabs. Figure 16 illustrates the load–time and load–deflection curves derived from these tests, encompassing the printed slab and cast slabs. Particularly, prior to reaching the peak point B, the load exhibited a similar trend to that of the standard disk in both the printed and cast slabs. However, in the printed slab, an unloading phase occurred before reaching peak point B. This phenomenon is attributed to the cracking of the outer shell in response to deflection. Also, in the case of the printed sample, a horizontal crack manifested due to the weakened bond points between layers. This observation is reflected in the DIC results, as depicted in Figure 17. Upon reaching the peak point (point B) in the load, a brief unloading event transpired, followed by the instantaneous failure of the interior of the test specimen.

Examining the strain evolution diagram of the printed slab, as illustrated in Figure 17, several noteworthy observations can be made. Before reaching point B in the load, there was a discernible concentration of strain distribution. However, it was at the moment of loading B that a pronounced strain concentration became evident, primarily situated at the specimen’s centre. As the loading process advanced, this concentration gradually expanded from the centre to encompass the entirety of the specimen, accompanied by a surge in strain levels from 0.07 to approximately 0.95. The load continued to ascend until it reached point C, at which juncture the crack propagated to both ends of the specimen, ultimately leading to complete specimen failure by the conclusion of the test. A similar pattern was discernible in another set of strain evolution diagrams, this time pertaining to cast slabs, as depicted in Figure 18. Interestingly, axial cracks attributed to the 3D printing process manifested in these cast slabs as well. This observation confirms that the failure of the specimens stemmed from a primary crack initiating at the loading point and subsequently generating additional cracks over time, extending along the course of loading. Furthermore, it establishes that beam specimens experience crack formation originating from stress concentration at the bottom, propagating towards the printed blocks.

### 3.2. Loading Stress versus Fractional Change in Resistivity (FCR) Relationship

Figure 19 provides an insightful comparison of the FCR and loading stress correlations between printed and cast samples. Both sets of data follow a similar trend and exhibit a degree of linearity. Polynomial regression functions were employed to capture the relationship between FCR and loading stress for the printed test specimens, demonstrating an approximately linear expression. The corresponding R-squared (R²) values for both sets of data were approximately 0.8. While these values offer valuable reference points, they may not be deemed precise enough for exact quantification.

The polynomial functions also exhibited a favourable fitting effect when applied to the scattered results from the cast specimens. Notably, at higher loading stress levels, both printed and cast specimens exhibited excellent piezoresistive responses, typically around 0.01 MPa. However, it is evident that the cast specimens displayed a greater degree of data dispersion between FCR and loading stress, particularly during the initial loading phase.

In contrast, the 3D-printed specimens demonstrated reduced data discretization when compared to their cast counterparts. Moreover, the determination coefficient (R²) was calculated to assess the degree of fit. Importantly, the 3D-printed test specimens did not exhibit significantly higher R² values than the cast test specimens. This observation suggests that, under flexible testing conditions, the 3D-printed specimens showcased a piezoresistive performance that closely parallels that of the cast specimens. Consequently, the repeatability and accuracy of the test results appeared to be similar between the two types of specimens [25].

Figure 20 illustrates the connection between flexural stress and FCR for both 3D-printed and cast slab specimens, and this relationship is modelled using polynomial functions. The correlation between these variables in both sets of samples is evident. However, when examining the relationship in the printed specimens, there is a noticeable level of scattering, and the R² value is approximately 0.82. Notably, no discernible regular pattern emerges in this scatter, which can be attributed to the presence of cracks in the printed specimens, as evidenced in Figure 17 and Figure 18. These cracks coincided with the locations where copper meshes served as electrodes. In stark contrast, the cast sample exhibited the best fit and minimal data scatter within this experiment, boasting an R² value of 0.96. This observation implies that the cementitious composite matrix in the cast specimens significantly enhances the repeatability and signal quality of piezoresistive performance, particularly in longer samples. In terms of the results, the printed slab sample achieved a maximum FCR of around 16%, while the cast slab demonstrates a robust stress-to-FCR correlation.

### 3.3. DIC Strain Measurement at the Central Point of the Specimen

After the fracture of the test piece, the strain gauge became inoperable, leading to the cessation of strain data recording beyond a certain point in the test. Consequently, the DIC calculation results, along with the strain gauge measurements obtained from the centre of the test piece prior to fracture, were analysed. These data are presented in Figure 21, where the red line represents the DIC measurement results, while the solid blue line represents the strain gauge measurements. It is important to note that due to the substantial computational load involved in DIC analysis, successful calculation was achieved for only one position within the sample. At the initiation of loading, the strain at the centre of the specimen generally exhibited linear characteristics, increasing steadily over time. However, as the load approached a specific point, strain concentration became evident in the specimen, leading to a rapid increase in strain. Figure 21 underscores the high correlation between the strain measured at the centre of the specimen using the DIC method and that obtained via the strain gauge under the flexure loading mode. This finding substantiates the feasibility of utilizing the DIC method for deformation measurements in concrete materials.

## 4. Conclusions

This study aimed to explore the mechanical, electrical, and piezoresistive characteristics of 3D-printed cementitious composites produced through extrusion printing. A comparative analysis was conducted against traditionally cast samples. Flexural tests were performed on both 3D-printed and cast specimens, including beams and slabs. The evolution of surface strain fields and electrical properties under identical loading conditions was investigated, employing the DIC method. Additionally, strain gauge measurements were employed to validate the suitability of the DIC method for concrete deformation assessment. The key findings can be summarized as follows:
Utilizing the DIC method, the displacement field on the surface of strengthened beam specimens is determined by processing collected images. This information, coupled with the strain field, facilitates the measurement of main crack morphology under the displacement field.In all directly loaded specimens, strain concentration initially emerges near the bottom of the loading end as the load increases. This concentration then rapidly extends toward the centre, culminating in a continuous cracking process until specimen failure occurs. In the case of 3D-printed samples subjected to flexural loading, strain concentration also initially occurs at the loading position of the specimen. As loading continues, cracks within the printed blocks propagate from the top centre to both ends, eventually penetrating the blocks axially.An analysis of FCR indicates that 3D-printed samples exhibit higher electrical responsiveness throughout the loading test, including higher peak values. However, at high loading stress levels, interference may arise from debris resulting from the fractured outer shell. Therefore, a more effective solution for 3D printing under such conditions warrants further investigation.The consistent measurement results between strain gauge data and DIC results confirm the reliability and suitability of the DIC method for measuring concrete deformation.

## Figures and Tables

**Figure 1 sensors-23-08486-f001:**
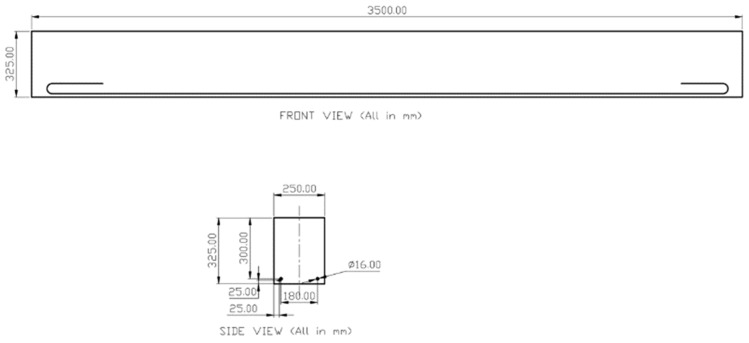
RC beams section.

**Figure 2 sensors-23-08486-f002:**
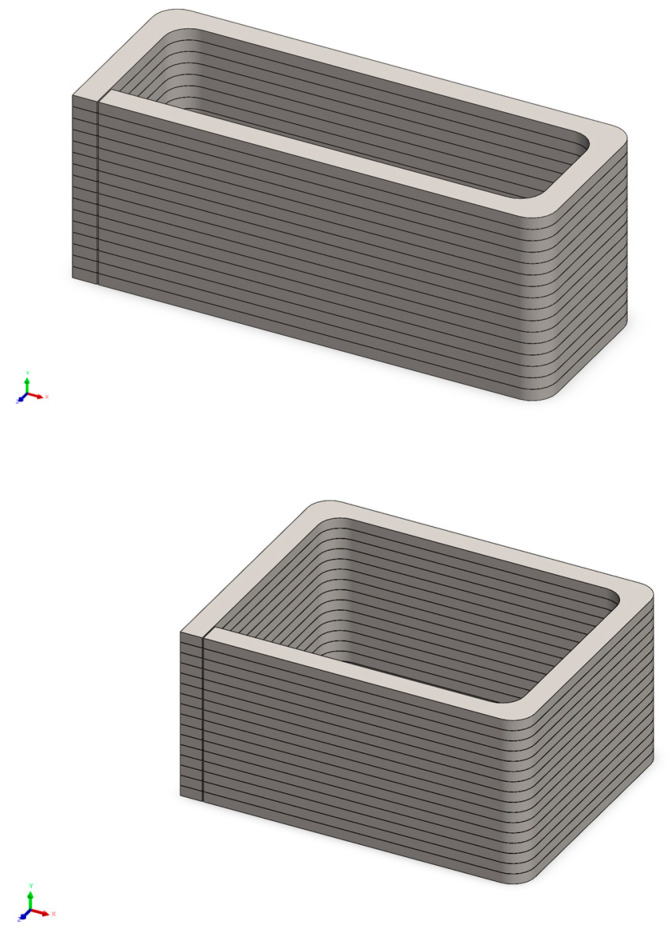
3D-printed blocks’ layouts.

**Figure 3 sensors-23-08486-f003:**
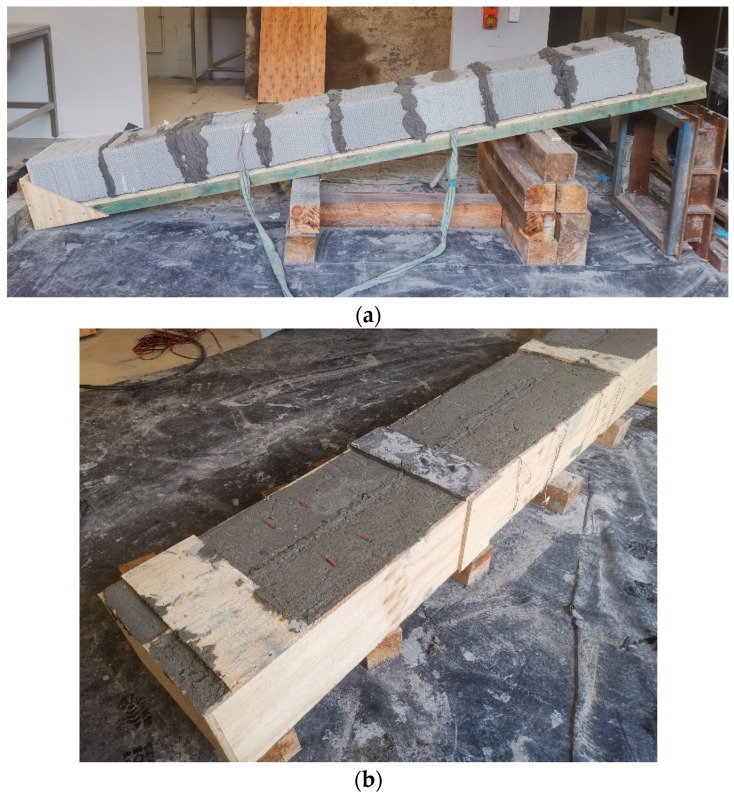
(**a**) 3D-printed blocks’ assembly sample and (**b**) mould cast sample.

**Figure 4 sensors-23-08486-f004:**
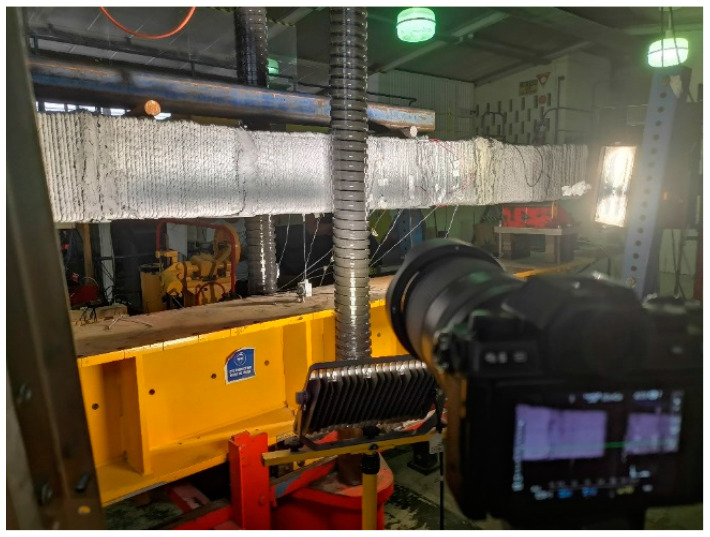
DIC system configuration.

**Figure 5 sensors-23-08486-f005:**
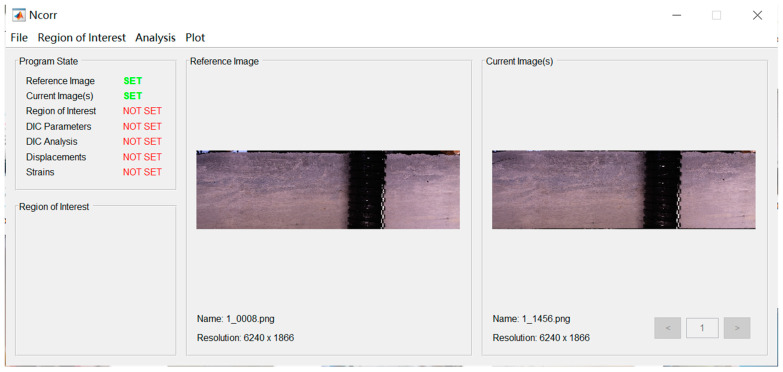
Interface of MultiDIC.

**Figure 6 sensors-23-08486-f006:**
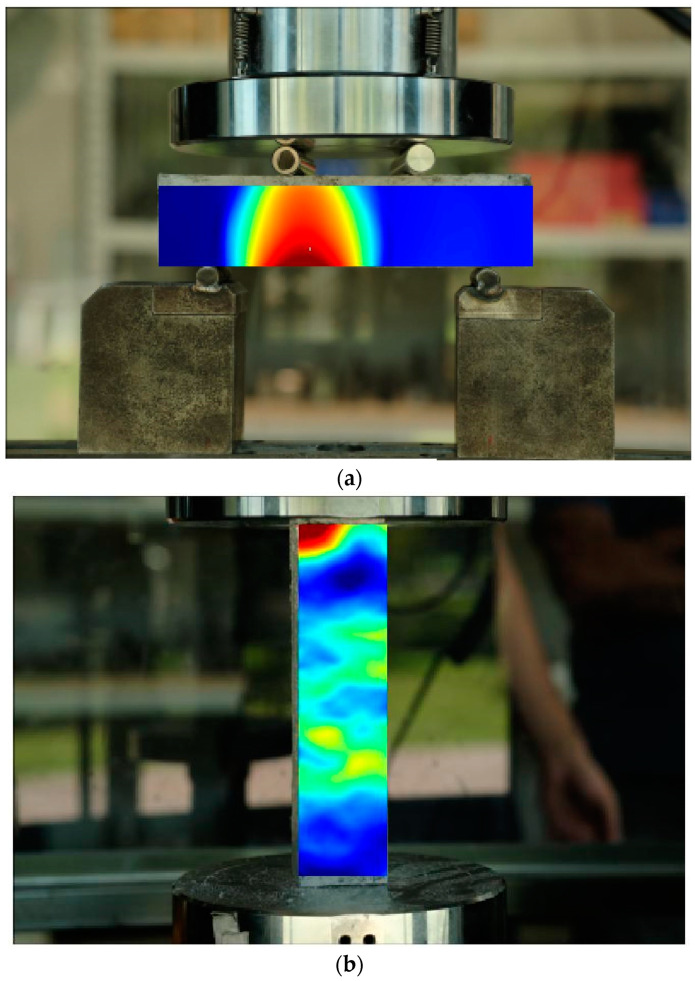
Typical DIC results of (**a**) flexural and (**b**) cycling piezoresistivity tests.

**Figure 7 sensors-23-08486-f007:**
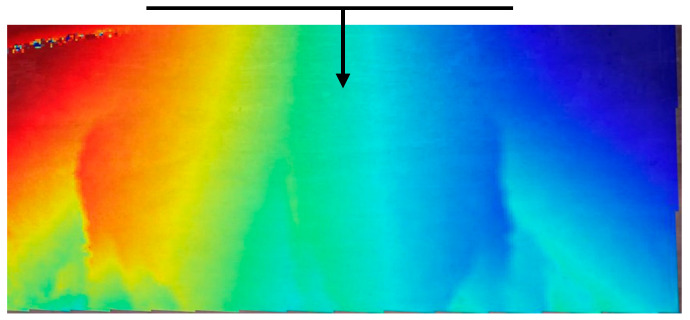
Displacement field of cast beam.

**Figure 8 sensors-23-08486-f008:**
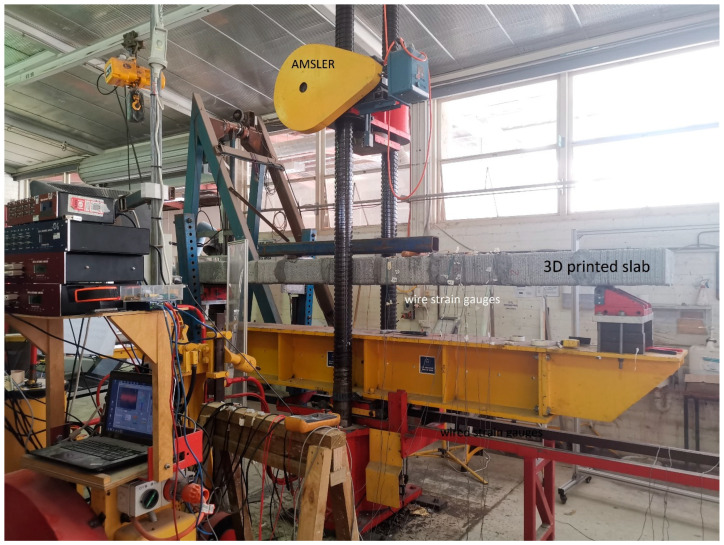
Configuration of AMSLER loading system.

**Figure 9 sensors-23-08486-f009:**
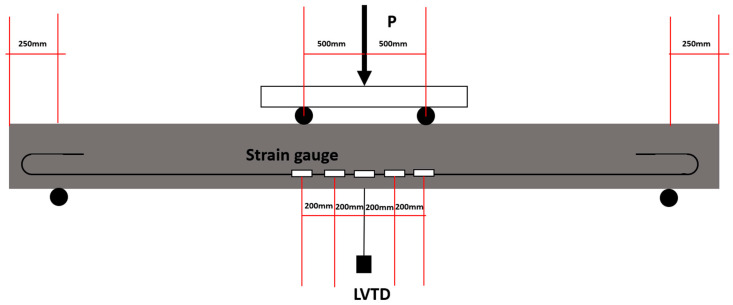
The setup for four-point bending test and strain gauges’ arrangement.

**Figure 10 sensors-23-08486-f010:**
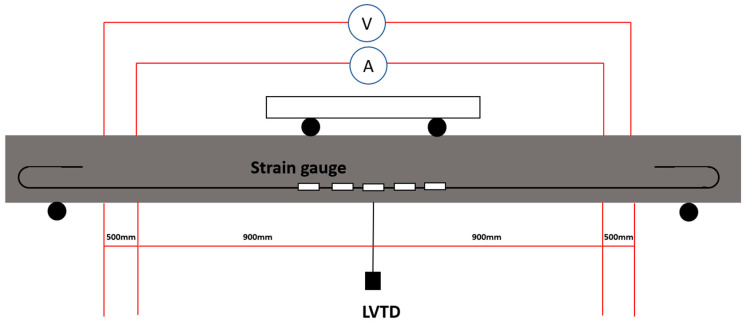
Schematic of copper meshes arrangement.

**Figure 11 sensors-23-08486-f011:**
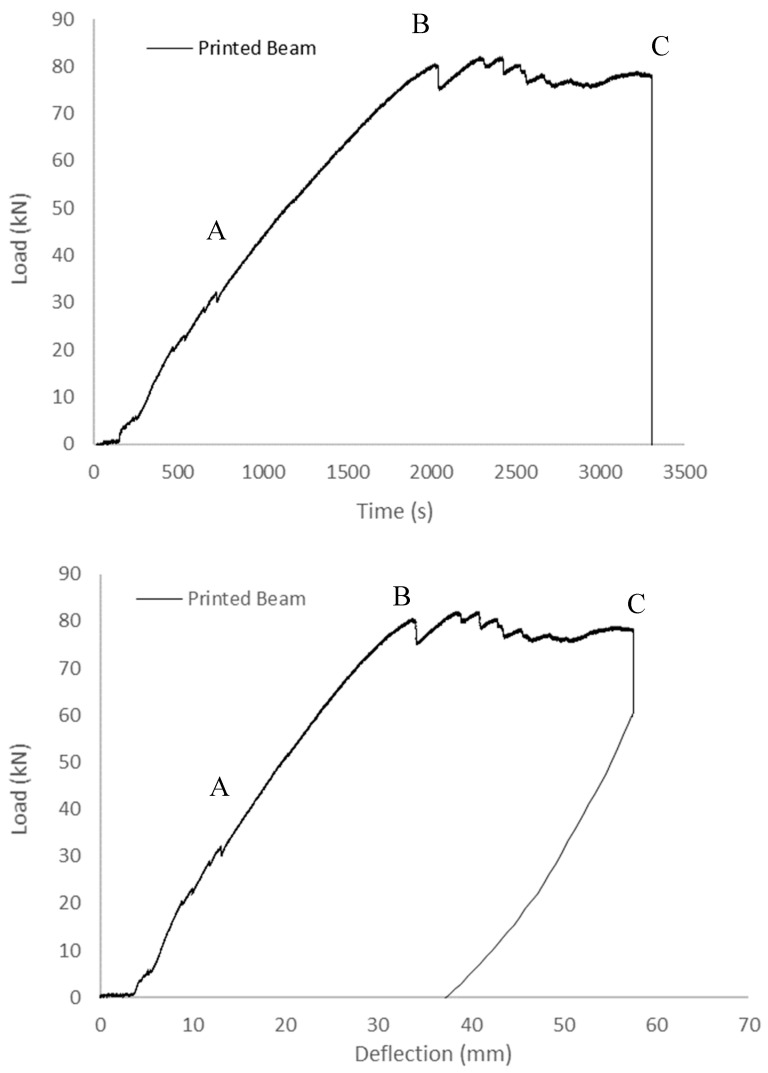
Load–time and load–deflection curves of printed beam (A: non-linear initial loading phase, B: yield point, and C: failure point).

**Figure 12 sensors-23-08486-f012:**
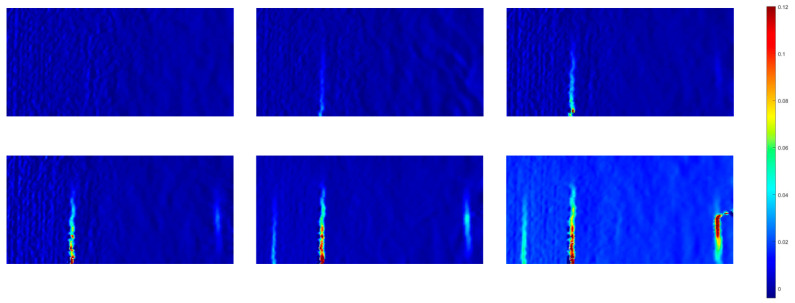
Evolution of strain field of the 3D-printed beam.

**Figure 13 sensors-23-08486-f013:**
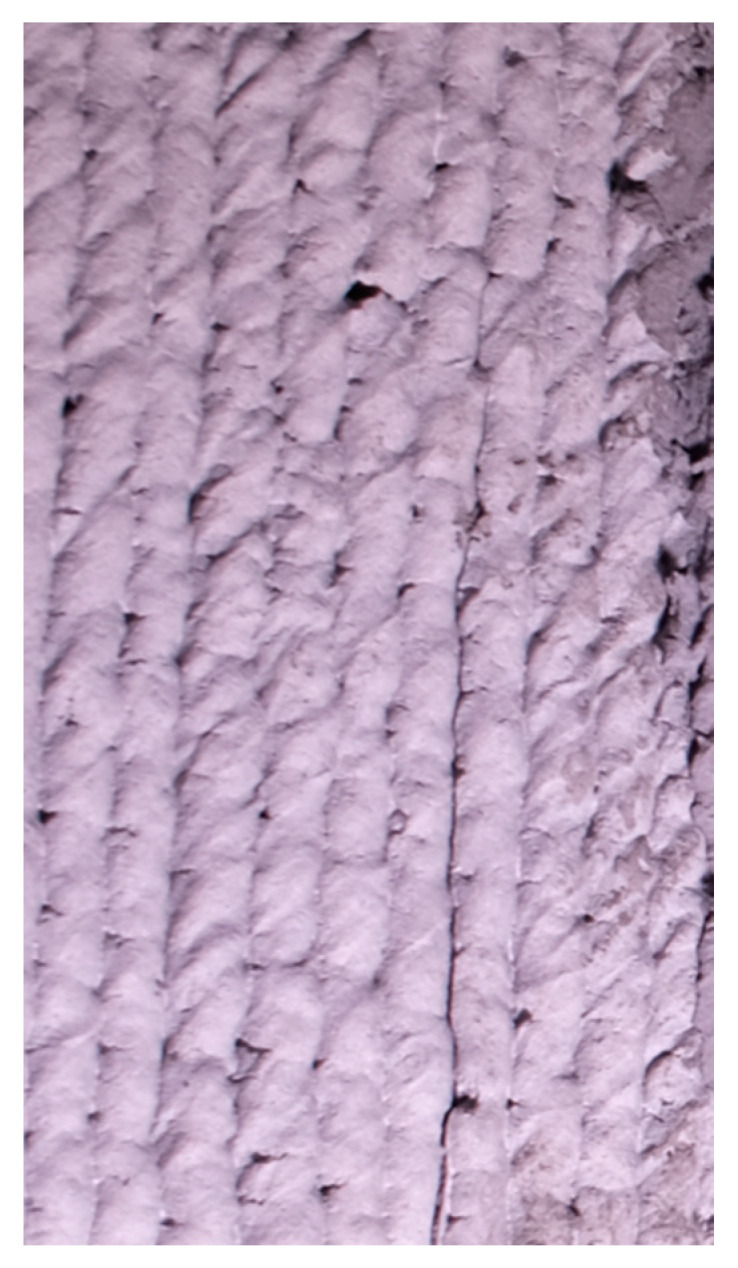
Zoomed in cracks.

**Figure 14 sensors-23-08486-f014:**
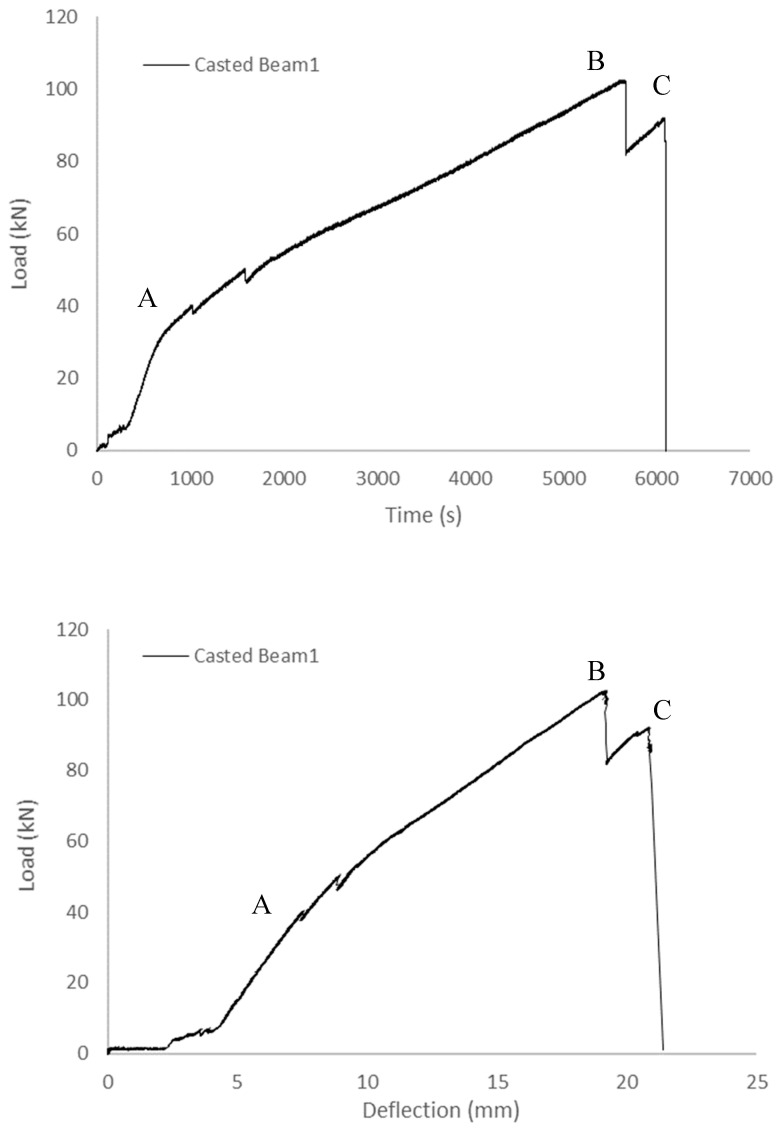
Load–time and load–deflection curves of the cast beams (A: non-linear initial loading phase, B: yield point, and C: failure point).

**Figure 15 sensors-23-08486-f015:**

Evolution of strain field of the cast beams.

**Figure 16 sensors-23-08486-f016:**
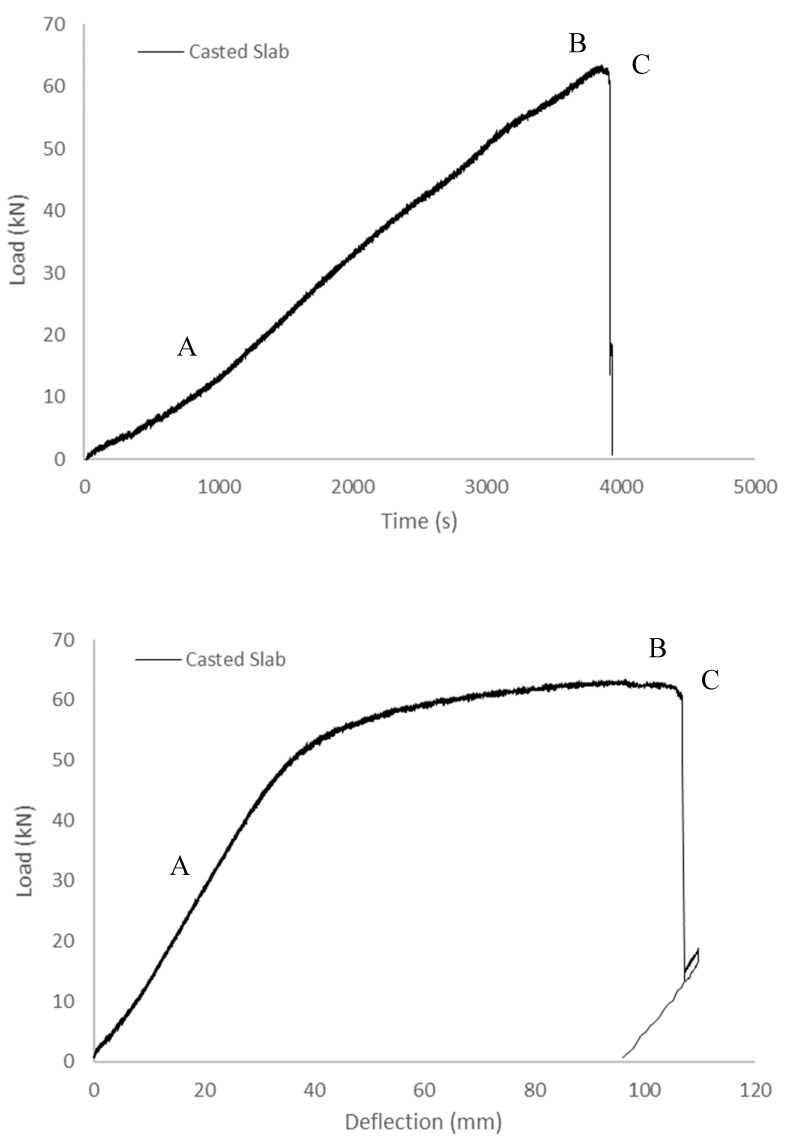
Load–time and load–deflection curves of the printed and cast slabs (A: non-linear initial loading phase, B: yield point, and C: failure point).

**Figure 17 sensors-23-08486-f017:**
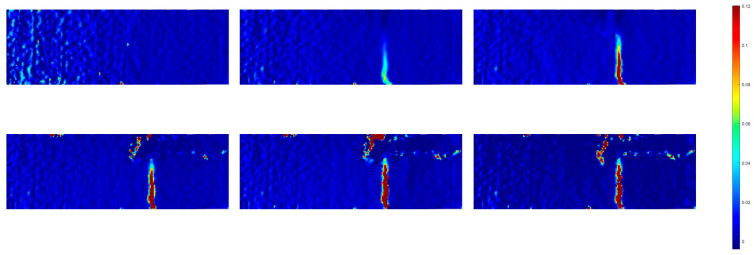
Evolution of strain field of the 3D-printed slab.

**Figure 18 sensors-23-08486-f018:**
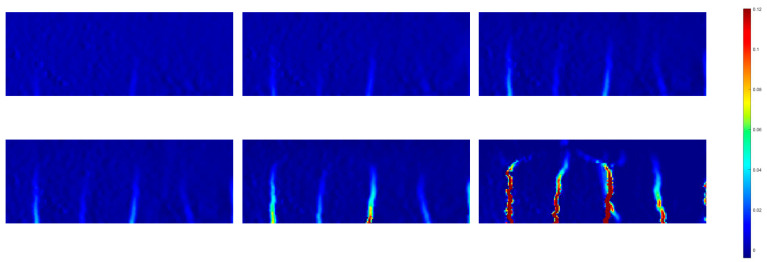
Evolution of strain field of the cast slab.

**Figure 19 sensors-23-08486-f019:**
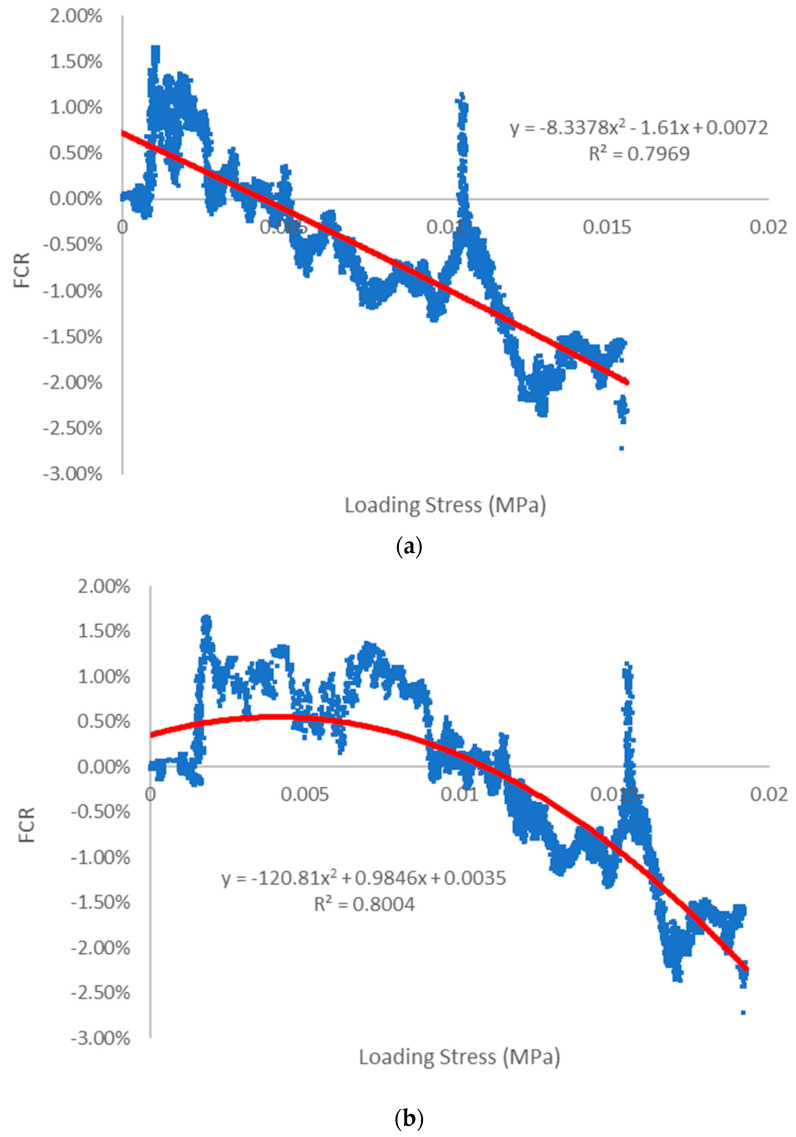
Loading stress versus FCR correlation for experimental reading (blue line) and fitted curved (red line) of (**a**) 3D-printed beam and (**b**) cast beam.

**Figure 20 sensors-23-08486-f020:**
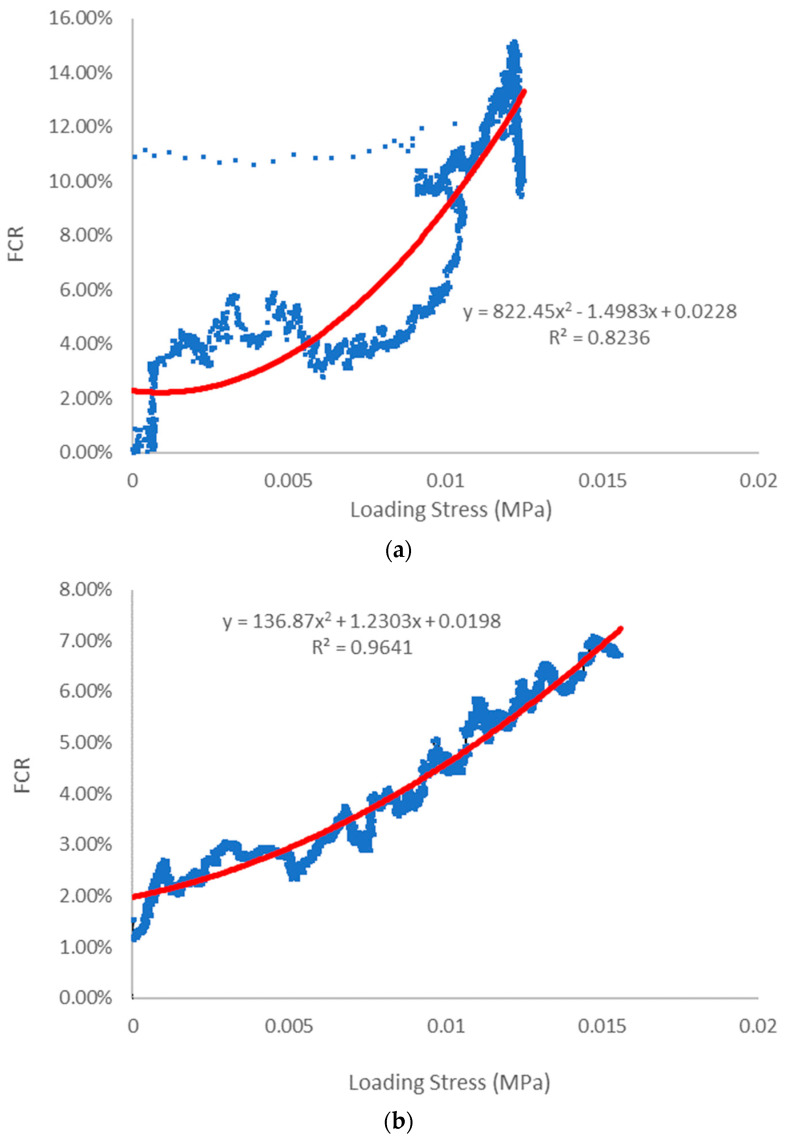
Loading stress versus FCR correlation for experimental reading (blue line) and fitted curved (red line) of (**a**) 3D-printed slab and (**b**) cast slab.

**Figure 21 sensors-23-08486-f021:**
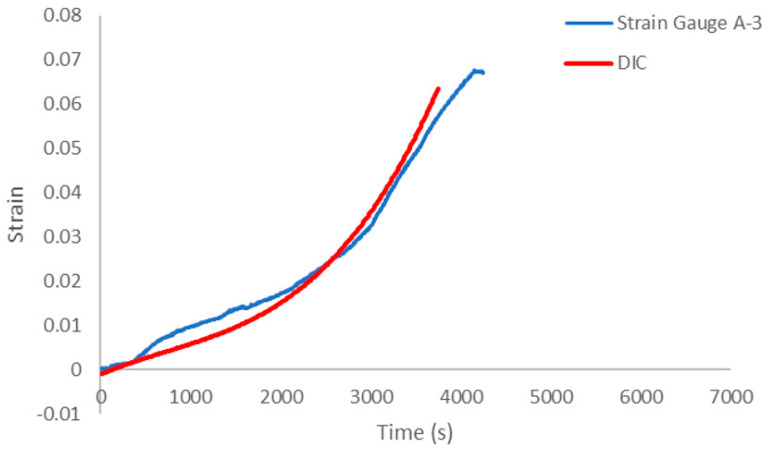
Comparison between strain determined from DIC analysis and by strain gauges.

**Table 1 sensors-23-08486-t001:** Chemical composition of general-purpose Portland cement.

	CaO	SiO_2_	Al_2_O_3_	Fe_2_O_3_	SO_3_	MgO	Na_2_O	Chloride
Cement	63.4	20.1	4.6	2.8	2.7	1.3	0.6	0.02

**Table 2 sensors-23-08486-t002:** Physical properties of general-purpose Portland cement.

	S.G.	Fineness Index	Normal Consistency	Setting Time Initial	Setting Time Final	Soundness	Loss of Ignition	Residue 45 μm Sieve
Cement	3.0 t/m^3^	390 m^2^/kg	27%	120 min	210 min	2 mm	3.80%	4.70%

**Table 3 sensors-23-08486-t003:** Chemical composition of GGBFS.

	FeO	SiO_2_	Al_2_O_3_	MgO	TiO_2_	MnO	Hydraulic Index
Slag	38%	32%	13%	5%	1.50%	0.50%	1.70%

**Table 4 sensors-23-08486-t004:** Physical properties of GGBFS.

	Bulk Density	Glass Content	Angle of Repose	Chloride Ion
Slag	850 kg/m^3^	>85%	35 degree	<0.025%

**Table 5 sensors-23-08486-t005:** Chemical composition of densified SF.

	Na_2_O	K_2_O	Alkali	Cl^−^	Anhydride	SO_3_
Silica fume	98%	0.33%	0.17%	0.40%	0.15%	0.83%

**Table 6 sensors-23-08486-t006:** Physical properties of densified SF.

	Bulk Density	Relative Density	Pozzolanic Activity at 7 Days	Control Mix Strength	Moisture Content	Loss of Ignition
Silica Fume	625 kg/m^3^	2.21	111%	31.3 MPa	1.10%	2.40%

**Table 7 sensors-23-08486-t007:** Properties of carbon fibre.

	Length	Diameter	Aspect Ratio	Density	Electrical Resistivity	Tensile Strength	Tensile Modulus	Carbon Content
CF	3 mm	0.007 mm	429	1.8 g/cm^3^	0.00155 Ωcm	4137 MPa	242 Gpa	95.00%

**Table 8 sensors-23-08486-t008:** Properties of activated carbon powder.

	Particle Size	Bulk Density	pH
ACP	200 mesh	0.38 g/cm^3^	3–5

**Table 9 sensors-23-08486-t009:** Mix composition.

	w/b Ratio	Cement	Slag	Silica Fume	Silica Sand	CF	ACP
Mix	0.325	1	1.2	0.11	0.836	0.7	0.25

## Data Availability

Some or all data, models, or code that support the findings of this study are available upon reasonable request from the corresponding author.

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
