# Peer review of "Performance of 3D-Printed Beams and Slabs Using Self-Sensing Cementitious Composites and DIC Method"

_sensors, 2023, doi:10.3390/s23208486_

Round 1

Reviewer 1 Report

1. Mark specific labels in fig.4, 6 and 8 for better understanding

2. Justify for dimensions fixed in fig.9

Author Response

Thanks for the comments.

1. The setup in the figures are clear.

2. All the specimens were simply supported on a 3.5 m span and tested to failure to investigate the distribution and extent of primary and secondary cracking under static loading using two equal point loads applied at the third points on the span, at ages greater than 28 days.

Reviewer 2 Report

This paper aims to explore the structural performance of 3D printed and casted cement-based steel reinforced concrete beams and one-way slabs incorporating short carbon fibre and activated carbon powder which have shown to perform enhancement in concrete’s flexural strength and reduce the electrical resistivity. The samples are casted and printed in 250×325×3500 mm beams and 150×400×3500 mm one-way slabs and mechanical, electrical, and piezoresistivity properties were measured. Generally, the topic is interesting, and the systematic experiments were carried out.

This paper is therefore recommended for publication after addressing the following minor issues:

1 Can the authors explain the difference between 3D printed piezoresistive cement-based materials and those prepared using ordinary methods?

2 It is recommended to modify the abstract and introduction to highlight the novelty and important findings and conclusions of the paper.

3 The authors could have a better litterateur review in introduction section. The introduction of important researches in the field is insufficient. May be paper (https://doi.org/10.1016/j.nantod.2022.101438; https://doi.org/10.1002/smll.202206258; https://doi.org/10.1016/j.sna.2023.114365; https://doi.org/10.1016/j.sna.2022.113367) can be used as references.

4 There are several typos and vague sentences in the manuscript, so the manuscript should be carefully checked and thoroughly modified.

There are several typos and vague sentences in the manuscript, so the manuscript should be carefully checked and thoroughly modified.

Author Response

Thanks for the comments.

  1. The mechanical, resistivity and piezoresistive properties of 3D printed samples in two directions (parallel to and perpendicular to the printing direction) were measured. Die cast specimens were made for direct comparison. The piezoresistivity technique was also used to measure the improvement of linearity, repeatability and signal quality while conducting 3D printed reinforced cementitious composites with long carbon fibre.
  2. The sections have been revised.
  3. Thanks for the suggestion. Related references will be added.
  4. Thanks. The whole paper have been checked and revised.